



# Validation Strategies for Deep Learning-Based Groundwater Level Time Series Prediction Using Exogenous Meteorological Input Features

Fabienne Doll[1], Tanja Liesch[1], Maria Wetzel[2], Stefan Kunz[2], and Stefan Broda[2]

[1]Institute of Applied Geosciences, Division of Hydrogeology, Karlsruhe Institute of Technology, Karlsruhe, Germany
[2]Federal Institute for Geosciences and Natural Resources, Berlin, Germany

**Correspondence:** Fabienne Doll (fabienne.doll@kit.edu)

**Abstract.** Due to the growing reliance on machine learning (ML) approaches for predicting groundwater levels (GWL), it is important to examine the methods used for performance estimation. A suitable performance estimation method provides the most accurate estimate of the accuracy the model would archive on completely unseen test data to provide a solid basis for model selection decisions. This paper investigates the suitability of different performance evaluation strategies, namely blocked
cross-validation (bl-CV), repeated out-of-sample validation (repOOS), and out-of-sample validation (OOS), for evaluating one-dimensional convolutional neural network (1D-CNN) models for predicting groundwater level (GWL) using exogenous meteorological input data. Unlike previous comparative studies, which mainly focused on autoregressive models, this work uses a non-autoregressive approach based on exogenous meteorological input features without incorporating past groundwater levels for groundwater level prediction. A dataset of 100 GWL time series was used to evaluate the performance of the different
validation methods. The study concludes that bl-CV provides the most representative performance estimates of actual model performance compared to the other two performance evaluation methods examined. The most commonly used OOS validation yielded the most uncertain performance estimate in this study. The results underscore the importance of carefully selecting a performance estimation strategy to ensure that model comparisons and adjustments are made on a reliable basis.

## 1 Introduction

The utilization of machine learning (ML) approaches to predict groundwater level (GWL) has increased significantly over the last two decades (Tao et al., 2022). The variety of prediction models ranges from simple methods such as tree-based models (e.g. Moghaddam et al., 2022) to more complex deep learning (DL) methods such as Artificial Neural Networks (ANN) (e.g. Derbela and Nouiri, 2020; Iqbal et al., 2020; Nair, 2016), Long-Short-Term-Memory Networks (LSTM) (e.g. Vu et al., 2021; Wunsch et al., 2021; Zhang et al., 2018; Gholizadeh et al., 2023) or Convolutional Neural Networks (CNN) (e.g. Wunsch et al.,
2021; Sun et al., 2019; Gomez et al., 2024). According to the review study by Tao et al. (2022), the most commonly used input features for GWL prediction are historical GWL (34%), precipitation (23%) and temperature (15%). Depending on the type of aquifer and the geographical location of the GWL observation well, other features such as sea level or groundwater abstraction rates are also used as input features (Tao et al., 2022).





In GWL prediction, we are faced with an increasing number of potential prediction models, a growing number of hyper-parameters for increasingly complex models, and a large number of possible input features. In order to find the right method for a particular application, it is necessary to compare the individual methods and to evaluate the results correctly. There is a general consensus on the usual approach to selecting an appropriate prediction method, described for example in Chollet (2021): i) Select a model, appropriate hyperparameters (e.g. learning rate, batch size, hidden size, number of epochs) and a set of promising input features. ii) Train the model on the training data set. iii) Evaluate the performance based on prediction error on the validation data set (validation error). To select an appropriate method, steps i) to iii) can be performed several times with different model configurations and compared on the basis of the validation error. Since fitting the model based on the validation error may result in overfitting to the validation data set, the final model should be retested on a completely unknown test data set to obtain the final model error and to get an idea of the generalization ability of the model (Chollet, 2021). Splitting the data into training, validation, and test sets ensures that the model is ultimately evaluated on completely unseen data. As model fitting and selection should therefore only based on training and validation data, a validation method is required that provides an accurate and robust estimate of the error the model would produce on completely unseen data. The choice of the optimal validation method depends primarily on the prediction problem and the type of data. The cross-validation (CV) is a very popular approach for independent and identicaly distributed (i.i.d) data (see Arlot and Celisse (2010)). In standard CV, the data is randomly shuffled and then divided into k-folds, with one fold used as the validation set and k-1 folds as the training set. The CV approach is particularly beneficial for small data sets, as the entire data can be used for model adaptation and validation (Chollet, 2021; Bergmeir et al., 2018; Arlot and Celisse, 2010). In context of predicting dependent data, particularly in autocorrelated time series data, CV is often considered problematic. This is because it disrupts the temporal dependency of the data (Arlot and Celisse, 2010), and there is a general consensus that future data should not be used to predict the past (Chollet, 2021). In addition, the aspect of potential non-stationarity (changing mean and variances over time or presence of long-term trends) is generally not taken into account in CV (Bergmeir and Benítez, 2012).

Therefore, out-of-sample validation (OOS) is preferred for time series forecasts. With standard OOS, the validation and test data sets are separated at the end of the time series (closer to the present) and the older data (further in the past) are used for training (Tashman, 2000). In contrast to CV, OOS only has one validation data set at the end of the time series for model adjustment and evaluation. Consequently, the advantage of CV, where multiple evaluations are performed on different validation data sets, cannot be utilized with OOS (Bergmeir et al., 2018). The OOS is also used most commonly to validate groundwater level predictions (Ahmadi et al., 2022).

Modifications to the standard CV method have been developed to take advantage of CV for validating time series prediction models. One example is the blocked cross-validation (bl-CV) (Snijders, 1988), in which the data are not randomly shuffled before being divided into individual folds. This preserves the temporal relationship of the data within the folds. Earlier studies from other domains have shown that CV methods are the most reliable performance estimators for autoregressive prediction of stationary time series [e.g.] (Bergmeir and Benítez, 2012; Bergmeir et al., 2014; Cerqueira et al., 2020). For the autoregressive forecasting of non-stationary time series, on the other hand, according to Cerqueira et al. (2020), the repeated out-of-sample validation (repOOS) method provided the most accurate estimates. In autoregressive time series prediction, the past values of





the time series are used for prediction of future values of the time series. In these studies, different model types were examined, including linear models, support vector regression, multilayer perceptrons, rule-based methods, and ensemble methods such as random forest models (Bergmeir and Benítez, 2012; Bergmeir et al., 2014, 2018; Cerqueira et al., 2020, 2017). To the best of the authors' knowledge, there is no study in the domain of groundwater level prediction research that systematically compares different methods for performance estimation. This research gap is addressed by the present work.

We demonstrate the applicability of k-fold bl-CV for GWL time series prediction and enable a direct comparison with simple OOS validation – the most commonly used method for performance estimation in GWL prediction (Ahmadi et al., 2022) – as well as with repeated OOS validation (repOOS), which has been identified as the most accurate method for non-stationary time series in the study by Cerqueira et al. (2020). Previous comparative studies have been limited to autoregressive forecasting approaches (e.g. Bergmeir and Benitez, 2011; Bergmeir and Benítez, 2012; Bergmeir et al., 2014, 2018; Cerqueira et al., 2020). In contrast, this study applies a non-autoregressive approach, which assumes that future groundwater levels are significantly influenced by past meteorological events – particularly precipitation and temperature. Accordingly, only meteorological input features are used. A practical advantage of this approach is the ability to make long-term predictions (e.g., for 10, 50, or 100 years) based on climate scenarios. A one-dimensional convolutional neural network (1D-CNN) is employed as prediction model – an architecture that has not been considered in any of the previous comparative studies and has proven to be reliable and robust for GWL prediction before (e.g Wunsch et al., 2021). By comparing OOS, bl-CV, and repOOS, this study contributes to methodological development in hydrogeological research by explicitly basing performance estimation on a real-world application in groundwater prediction. The investigation is based on 100 real GWL time series, including both stationary (constant mean and variance over time) and non-stationary (changing mean and variance over time) conditions.

The following research questions are to be answered in the course of this work:

- Are there differences in the quality of the performance estimates of the three tested validation methods in the GWL time series prediction?

- Which validation method provides the most robust performance estimate in general, for stationary and for non-stationary groundwater conditions?

- Do our results agree with the results of previous studies in the field of autoregressive time series forecasting in other disciplines, or are there differences?

## 2 Theory and Background

### 2.1 Validation methods

#### 2.1.1 Out-of-sample validation (OOS)

The standard type of OOS refers to a validation process in which the data set is chronologically divided into subsets. After separating the test data, which often represents the most recent data, the older part of the remaining data is used as the training




data set, while the newer part is used for validation. With the OOS, the validation error (performance estimate) of the individual validation data set at the end of the time series is obtained as the performance estimate (see Fig. 1a). The training and validation subsets are often divided according to percentage of the total data set (e.g. 80% training data and 20% validation data). A division of subsets by calendar year, to account for seasonal behavior in environmental time series, is also common practice (e.g. Wunsch et al., 2021; Gomez et al., 2024). The OOS approach is most straightforward, as it comes closest to the actual application of time series forecasting, namely predicting the future from the past. In addition, this approach takes into account the assumption that the future is linked to the past and that past processes therefore influence the future. The weakness of OOS is that the validation data set may differ in its properties from those of the training and test datasets, and the resulting validation error could be unrepresentative (Bergmeir and Benítez, 2012).

For a more robust performance estimate, it is recommended to apply the OOS strategy to multiple validation periods (Tashman, 2000). One possible approach for this is the repOOS approach, in which the OOS validation is performed multiple times with different, possibly overlapping training and validation periods (Cerqueira et al., 2017, 2020). This is done by randomly selecting a split point from an available split window within the time series. A fixed fraction (e.g. 60%) of the data before this split point serves as the training data set and a fixed fraction (e.g. 10%) after this point serves as the validation data set. Thus, the length of the available split window is limited by the size of the training and validation data (Cerqueira et al., 2017, 2020). The final validation error of the repOOS is the mean of the validation errors of all repetitions. The splitting scheme for one repetition of repOOS validation is illustrated in Figure 1b.

### 2.1.2 K-fold blocked cross-validation (bl-CV)

When using the standard k-fold CV, the data is split into k equal-sized, non-overlapping parts (folds). After splitting into k folds, k-1 folds are used to train the model and one fold serves as validation data. Typically, the data is randomly shuffled before splitting into folds. For time series prediction, random shuffling of the data is often considered problematic as it can break the temporal dependency of the data. We describe this problem in more detail in the next section 2.2.

Alternatively Snijders (1988) used k-fold CV, where the data is not randomly shuffled before being divided into folds. Without random shuffling, the temporal relationship within the folds is preserved, only the temporal relationship between the folds is interrupted. This method is referred to in the literature as blocked cross-validation (bl-CV) (e.g. Bergmeir and Benítez, 2012; Cerqueira et al., 2020). An example of a k-fold bl-CV (k=5) is shown in Figure 1c. Regardless of whether the data were shuffled before splitting or not, a k-fold CV results in k validation errors, one for each split. To obtain a final performance estimate, the mean of the k validation errors is calculated. The combination of the k validation errors usually gives a more robust performance estimate than the individual validation error from the OOS described in the section 2.1.1.





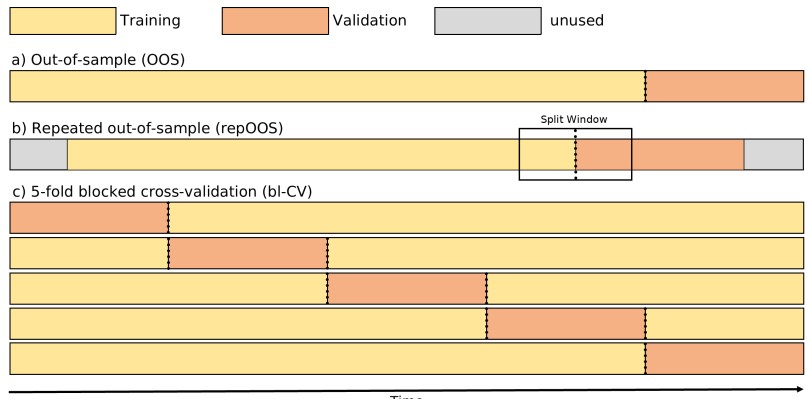

**Figure 1.** Schematic representation of OOS validation split (a), the splitting scheme of one repetition repOOS (b) and k-fold bl-CV (k=5) (c).

## 2.2 Cross-validation in time teries prediction

### 2.2.1 Autoregressive time series prediction

A time series is a sequential series of numerical values measured over a certain period of time:

$$\mathbf{Y} = \{y_1, y_2, y_3, \ldots, y_n\}, \quad y_t \in \mathbb{R} \tag{1}$$

Here, $y_t$ is the value of $\mathbf{Y}$ at time $t$, and $n$ is the total length of $\mathbf{Y}$. Time series forecasting refers to the problem of predicting a future value from a time series $\mathbf{Y}$ based on a given number of past observations (lags). In the case of autoregressive modeling, a regression is performed to predict the next value of $\mathbf{Y}$ using the previous p lags of $\mathbf{Y}$. Given a fixed number of $p$ lags and a forecast horizon $h = 1$, the time series is transformed as follows:

$$\mathbf{X}_{\mathrm{AR}} = \begin{bmatrix} y_1 & y_2 & \cdots & y_p \\ \vdots & \vdots & \vdots & \vdots \\ y_{t-p} & y_{t-p+1} & \cdots & y_{t-1} \\ \vdots & \vdots & \vdots & \vdots \\ y_{n-p} & y_{n-p+1} & \cdots & y_{n-1} \end{bmatrix} \quad \mathbf{y} = \begin{bmatrix} y_{p+1} \\ \vdots \\ y_t \\ \vdots \\ y_n \end{bmatrix} \tag{2}$$

Each row of the input feature matrix $\mathbf{X}_{\mathrm{AR}}$ represents a input feature vector formed by the past p lags of $\mathbf{Y}$, while the target vector $\mathbf{y}$ contains the corresponding target value, which is the next observed value in the time series $\mathbf{Y}$. For instance the first row of the feature matrix $\mathbf{X}_{\mathrm{AR}}$ are therefore the input features $\{y_1, y_2, \ldots, y_p\}$ and the corresponding taget value $y_{p+1}$ is in the first row of the target vector. In this form, input feature matrix $\mathbf{X}_{\mathrm{AR}}$ and target vector $\mathbf{y}$ can be used for the regression to find a suitable regression function, mapping the $p$-dimensional input feature vectors to their respective target values.

It should be noted that this type of time series embedding leads to large overlaps between the rows of $\mathbf{X}_{\mathrm{AR}}$ and $\mathbf{y}$, which introduces statistical dependencies between the training and validation data. In particular, a target value $y_t$ in the validation





data set may already appear in the lagged input of the training data set. This violates the assumption of statistical independence required in traditional CV (Arlot and Celisse, 2010; Bergmeir and Benítez, 2012) and can lead to a distorted performance assessment, since the model already has access to target values from the validation data set during training.

This overlap problem is mitigated by modified CV approaches, such as those proposed in McQuarrie and Tsai (1998), or by hv-blocked CV presented by Racine (2000). The bl-CV described in section 2.1.2 is also suitable for this dependent setting, as the overlap only occurs at the boundaries of the training and validation data. According to Bergmeir et al. (2018), however, random shuffling can also be performed for autoregressive predictions as long as the model residuals remain serially uncorrelated, which is the case if the model is well fitted to the data.

### 2.2.2 Time series prediction with exogenous features

In addition to the purely autoregressive approach described in section 2.2.1, feature sequences of $p$ lags from other time series can be included (exogenous input features). The inclusion of exogenous input features can improve the accuracy of the forecast, as the target variable may be correlated with other influencing factors (Castilho, 2020).

It is also possible to discard the autoregressive approach and consider only $p$ lags of exogenous input features. Given an exogenous time series $\mathbf{X}$, which has the same length as the target time series $\mathbf{Y}$ (see equation (1)):

$$\mathbf{X} = \{x_1, x_2, x_3, \ldots, x_n\}, \quad x_t \in \mathbb{R} \tag{3}$$

where $x_t$ is the value of $\mathbf{X}$ at time $t$, we can construct the following input feature matrix $\mathbf{X}_{\text{EX}}$ and the corresponding target vector $\mathbf{y}$, assuming a fixed number of $p$ lags and a prediction horizon of h=1:

$$\mathbf{X}_{\text{EX}} = \begin{bmatrix} x_1 & x_2 & \cdots & x_p \\ \vdots & \vdots & \vdots & \vdots \\ x_{t-p} & x_{t-p+1} & \cdots & x_{t-1} \\ \vdots & \vdots & \vdots & \vdots \\ x_{n-p} & x_{n-p+1} & \cdots & x_{n-1} \end{bmatrix} \quad \mathbf{y} = \begin{bmatrix} y_{p+1} \\ \vdots \\ y_t \\ \vdots \\ y_n \end{bmatrix} \tag{4}$$

Contrary to the matrix $\mathbf{X}_{\text{AR}}$ described in (2) each row of the new input feature matrix $\mathbf{X}_{\text{EX}}$ contains a input feature vector formed by the past $p$ lags of the exogenous time series $\mathbf{X}$. The target vector $\mathbf{y}$ remains the same as in the autoregressive case, containing the corresponding target value, which is the next observed value in the time series $\mathbf{Y}$. For instance the first row of the input feature matrix $\mathbf{X}_{\text{EX}}$ are therefore the input features $\{x_1, x_2, \ldots, x_p\}$ and the corresponding target value $y_{p+1}$ is in the first row of the target vector $\mathbf{y}$. While the structure of $\mathbf{X}_{\text{EX}}$ differs from the autoregressive input feature matrix $\mathbf{X}_{\text{AR}}$, the target vector $\mathbf{y}$ is unaffected. As in the autoregressive case, the input feature matrix $\mathbf{X}_{\text{EX}}$ and the target vector $\mathbf{y}$ can be used for the regression to find a suitable regression function that maps the $p$-dimensional input feature vectors to their respective target values.

The exogenous approach uses external time series for forecasting, unlike the autoregressive approach which uses past values of the target variable as input features. This approach does not rely on potential autocorrelations within the target series, but on





cross-correlations between the target variable and one or more exogenous time series. Because input features and target values originate from distinct time series, the exogenous approach is less constrained by validation techniques than the autoregressive approach. As can be seen in equation (4), the target values remain unique, with only consecutive values of the exogenous input features potentially appearing multiple times.

## 3 Experimental design

### 3.1 Target time series

The GWL target time series originate from the State Office for the Environment Brandenburg (LfU), publicly available on the Water Information Platform of the federal state Brandenburg. Initially, 809 GWL time series covering the period from at least January 1, 1990, to December 28, 2020, were considered. Preprocessing involved aggregating on a weekly basis (Monday), applying linear interpolation for measurement gaps of up to four weeks, and imputing larger gaps (up to 52 weeks) using an iterative imputer based on Bayesian Ridge from scikit-learn (Pedregosa et al., 2011).

In order to map different groundwater level dynamics in the data set, the time series were selected according to their stationary or non-stationary behavior. A time series is considered weakly stationary if its mean and variance remain constant over time and its autocovariance depends only on the time difference. Weakly stationary time series thus represent stable systems. In contrast, non-stationary time series indicate changing (time-dependent) processes, for example due to long-term trends or anthropogenic influences. All 809 time series were tested using the augmented Dickey-Fuller Unit Root (ADF) and Kwiatkowski-Phillips-Schmidt-Shin (KPSS) tests with default settings from the Python module statsmodels (Seabold and Perktold, 2010). A time series was classified as weakly stationary only if both tests confirmed stationarity. Otherwise, it was labeled as non-stationary. This analysis resulted in 294 time series being identified as weakly stationary, while 515 were classified as non-stationary. In the following, time series identified as weakly stationary will be referred to as stationary time series.

To reduce the computational effort, a subset of 100 GWL time series - consisting of 50 stationary and 50 non-stationary time series - was randomly selected (see Fig. 2) (Doll et al., 2025). Selecting equal numbers of both stationary and non-stationary groundwater level time series ensures that the data set includes time series with both constant means and variances and time series with changing means and variances over time, as well as long-term trends. Using this method, we were able to ensure a balanced selection of time series that reflects the full range of groundwater level variability in our data set.




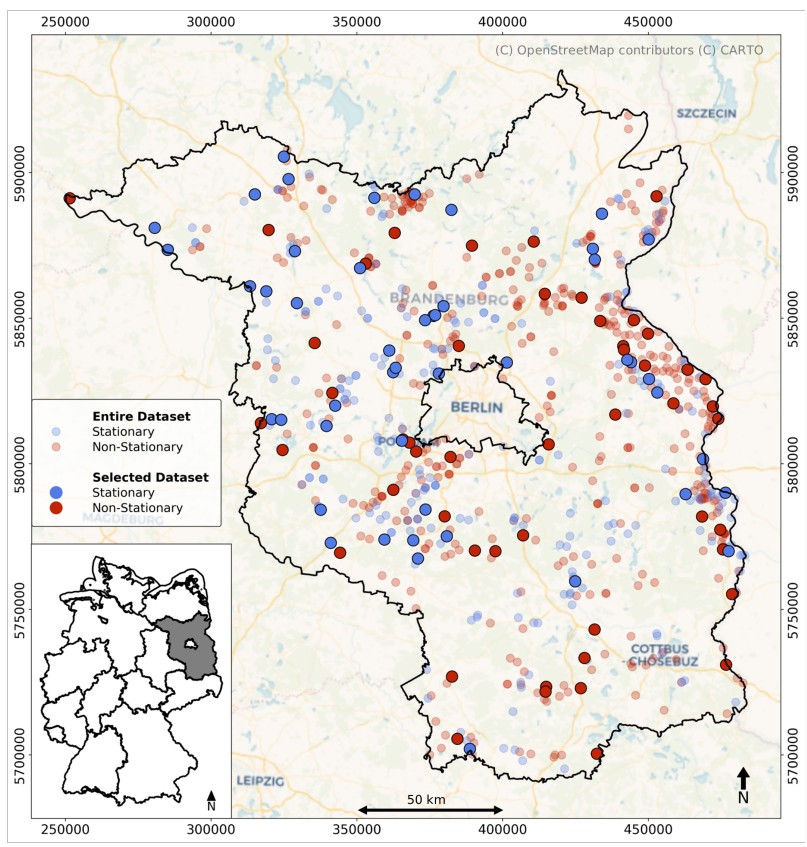

**Figure 2.** Map of the monitoring wells in the initially data set (809 monitoring wells) (semi-transparent colors) and the selected 100 monitoring wells used in this study (non-transparent colors). Colors indicate whether the time series are stationary (blue) or non-stationary (red).

### 3.2 Input time series

The meteorological parameters precipitation sum (P), mean air temperature (T), minimum air temperature (Tmin), maximum air temperature (Tmax) and relative humidity (rH) were used as exogenous features for the GWL forecasts. These were obtained from the publicly accessible HYRAS-DE v.5.0 dataset of the German Weather Service (DWD) (Rauthe et al., 2013; Razafimaharo et al., 2020). The HYRAS-DE v.5.0 dataset is a 5 x 5 km$^2$ raster dataset based on meteorological observations from weather monitoring sites around Germany. For each of the selected groundwater (GW) monitoring sites, the daily HYRAS data were extracted and aggregated to weekly values (Monday). As an additional input parameter, we used a sinosoidal curve
fitted to the mean air temperature (Tsin) to capture possible seasonal patterns (Wunsch et al., 2021).

### 3.3 Prediction task and model design

We applied a weekly sequence-to-value forecast, where a future groundwater level (forecast horizon h = 1 week) was predicted based on a sequence of p-lags (p = 52 weeks) from the exogenous meteorologacal input time series (this form of data embedding



was explained in section 2.2.2). Since we have selected weekly input and target time series from 1 January 1990 to 28 December
2020 (section 3.1) and an input sequence length of 52 weeks, the predictable GWL time series begins on 31 December 1990
(first predictable value) and ends on 28 December 2020 (last value of the time series). The feature and target data were
scaled to a range between 0 and 1 (MinMaxScaler; scikit-learn (Pedregosa et al., 2011)) before each training run based on the
corresponding training data. A 1D-CNN with keras (Chollet, 2015) was implemented for the prediction. The model architecture
and the associated hyperparameters are shown in Figure 3.

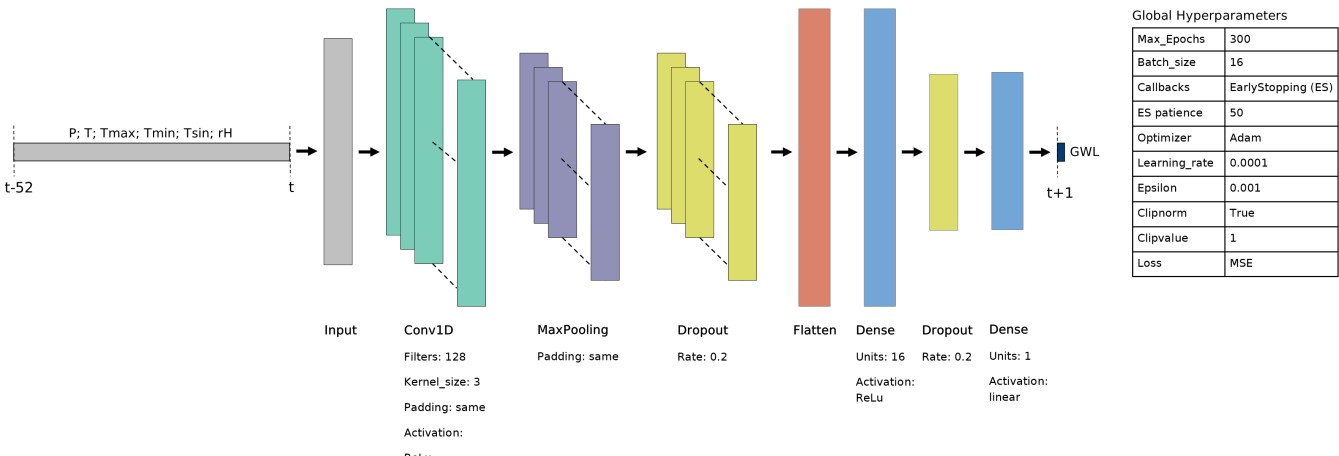

**Figure 3.** The model structure and the combination of the hyperparameters of the 1D-CNN model.

## 3.4  Training, validation and testing strategy

The objective was to assess the suitability of the three validation methods (bl-CV, repOOS, OOS) to provide the most accurate
performance estimate of the actual model performance based on the unseen test data. To create this test data set, the most
recent 20% of the measurements from each GWL time series were separated from the rest of the data. This portion is referred
as the out-set (Bergmeir et al., 2018). The separation of the test data was based on the standard OOS principle (section 2.1.1).
We chose this approach for separating the test data because our experiment was designed to simulate a real-world application
where the most recent data (representative of future data) are not available during model development and parameter tuning
based on the training and validation data.

The remaining 80% of the data was used to adapt the training and to validate the 1D-CNN. This data formed the so-called
in-set (Bergmeir et al., 2018). Within this in-set, the model was trained and validated in three different ways:

– OOS (Fig. 1a):    The model was trained and validated once with OOS training and validation split.

– repOOS (Fig. 1b):    The model was repeatedly trained and validated using an out-of-sample training and validation
split, controlled by the random choice of the split point in the split window. For the repOOS procedure, 60% of the in-set
was selected as training data and the following 10% as validation data, according to Cerqueira et al. (2020). The available





split window thus had a length of 30%, starting after 60% and ending at 90% of the length of the in-set. The number of
repetitions performed for this study was nrep = 5.

– bl-CV (Fig. 1c):   The model went through the k-folds of the k-fold bl-CV. The number of folds chosen for this study
was k = 5 in order to keep the validation periods as long as possible and to limit the computing effort.

In each training and validation split, an early stopping callback was set to determine the optimal number of training epochs
based on the validation data and to avoid overfitting. In this way, each training and validation split provides both a prediction
of the validation dataset and an estimate of the optimal number of training epochs.

Since the model performance depends on the initialization of the model weights, an ensemble of 10 independent 1D-CNN
models was implemented. Each model in the ensemble was initialized separately with a fixed random seed to ensure repro-
ducibility. This resulted in five validation predictions and five optimal epoch values for the repOOS and k-fold bl-CV and in
one validation prediction and one optimal epoch value for the OOS for each of the 10 ensemble members. To consolidate the
results of the 10 ensemble members, the median prediction and epoch number was calculated. The exact procedure is described
in detail below:

**Procedure for OOS**

– For the training and validation step, the 10 independently initialized models provided:

– 10 predictions of the validation dataset
– 10 optimal number of training epochs

– These 10 validation predictions and epoch numbers were combined into:

– A single median prediction of the validation dataset
– A single median optimal number of epochs (`Epochs_OOS`)

– The root mean squared error (RMSE) between the median validation prediction and the actual measured GWL was
calculated ($RMSE_{in}\_OOS$).

**Procedure for repOOS (nrep = 5)**

– The 10 initializations were built for each repetition, resulting in:

– 10 predictions of the validation dataset
– 10 optimal numbers of training epochs
– These 10 validation predictions and epoch numbers were combined into:





- • A single median prediction of the validation dataset

- • A single median optimal number of epochs

- • The RMSE between the median validation prediction and the actual measured GWL was calculated

- – After completing the five repetitions of repOOS, this gives:

- – Five RMSE (one per repetition)

- – Five estimates of the optimal number of epochs (one per repetition)

- – Finally, the mean of these five values was computed to obtain:

- – The final repOOS in-set error ($\texttt{RMSE}_{\texttt{in\_repOOS}}$)

- – The final repOOS epoch estimate ($\texttt{Epochs\_repOOS}$)

**Procedure for k-fold bl-CV (k = 5)**

- – The 10 initializations were built for each fold, resulting in:

- – 10 predictions of the validation dataset

- – 10 optimal numbers of training epochs

- – These 10 validation predictions and epoch numbers were combined into:

- • A single median prediction of the validation dataset

- • A single median optimal number of epochs

- • The RMSE between the median validation prediction and the actual measured GWL was calculated

- – After completing the 5-fold bl-CV, this gives:

- – Five RMSE (one per fold)

- – Five estimates of the optimal number of epochs (one per fold)

- – Finally, the mean of these five values was computed to obtain:

- – The final bl-CV in-set error ($\texttt{RMSE}_{\texttt{in\_bl-CV}}$)

- – The final bl-CV in-set epoch estimate ($\texttt{Epochs\_bl-CV}$)

It should be noted that in this study the training and validation data split of the last fold (Fold 5) of the 5-fold bl-CV
corresponds exactly to the data split used in OOS (compare Fig. 1 a and c). Therefore, the validation results from the last fold
of the 5-fold bl-CV and those of the OOS are the same. The results of bl-CV include those of OOS (Fold 5) and the other four
folds (Fold 1-4).




After validation at the in-set, the model is retrained with all the data from the in-set and the out-set is predicted. The number of training epochs in the out-set is determined by the optimum number of epochs determined during the corresponding validation of the in-set (`Epochs_OOS`, `Epochs_repOOS`, `Epochs_bl-CV`). The retraining and out-set prediction was also repeated for each of the 10 different initializations, and finally the median of the 10 ensemble predictions was calculated to obtain the final out-set prediction. Consequently, three predictions of the out-set are created. This prediction of the out-set allows the calculation of the RMSE between measured and predicted GWL of the out-set ($RMSE_{out}\_OOS$, $RMSE_{out}\_repOOS$, $RMSE_{out}\_bl-CV$) (see Fig. 4).

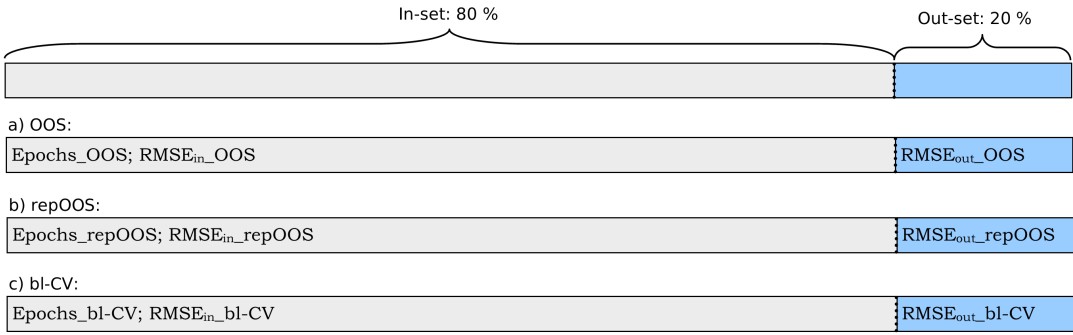

**Figure 4.** Illustration of the validation and testing process: Based on the validation methods in the in-set, the optimal number of training epochs (`Epochs`) and the model error in the in-set ($RMSE_{in}$) are determined. The model is then retrained with all the data from the in-set up to the number of training epochs determined by the respective validation method. The out-set is predicted and the prediction error ($RMSE_{out}$) is determined.

### 3.5 Performance estimation measurement

As described in section 3.4, each validation procedure leads to a performance estimate $RMSE_{in}$, which represents the prediction error in the in-set. The corresponding predictions of the out-set lead to the $RMSE_{out}$, which represents the actual error of the model on the unseen out-set. To compare $RMSE_{in}$ and $RMSE_{out}$, two addiational metrics were calculated following (Bergmeir et al., 2018): absolute predictive accuracy error (APAE) and predictive accuracy error (PAE)

$$\mathbf{APAE} = |RMSE_{in} - RMSE_{out}| \tag{5}$$

**APAE** can be used to determine the absolute error between the ($RMSE_{in}$) and ($RMSE_{out}$). The smaller the **APAE**, the better the performance estimation of the corresponding validation method.

$$\mathbf{PAE} = RMSE_{in} - RMSE_{out} \tag{6}$$

**PAE** shows whether the ($RMSE_{in}$) of the corresponding validation method overestimates or underestimates the ($RMSE_{out}$) (bias of the performance estimate). A positive **PAE** means the validation method overestimated the true error (pessimistic estimate). A negative **PAE** indicates that the validation method underestimated the true error (optimistic estimate).

As we predicted groundwater levels in metres above sea leve (m a.s.l.), the RMSE, APAE, and PAE are in meters.





# 4   Results and discussion

Upon training, validating and testing the 1D-CNN for each time series according to the strategy described in chapter 3.4,

one APAE and one PAE per time series were calculated for each of the three validation methods (bl-CV, repOOS, OOS). The

obtained APAE and PAE are presented visually in Figure 5 as boxplots and numerically in Table A1 differentiated by validation

method. The results were evaluated for all 100 predicted GWL time series and separately for stationary and non-stationary time

series. An overview of the resulting $RMSE_{in}$ and $RMSE_{out}$ is provided in Table A2.

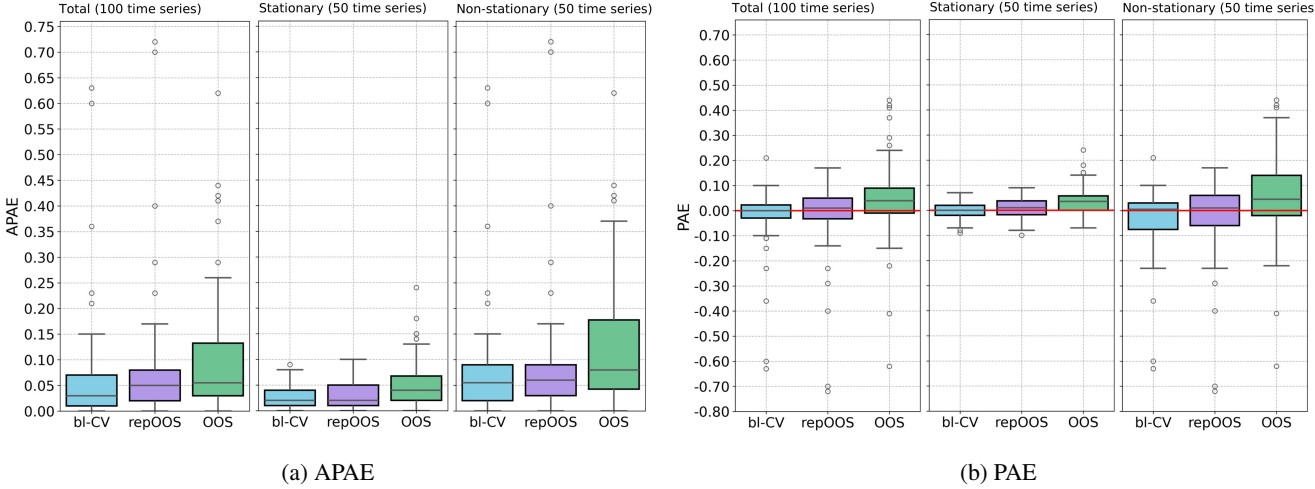

(a) APAE                                        (b) PAE

**Figure 5.** a) Achieved APAE per validation method across all 100 GWL time series (left), only for the 50 stationary time series (center), only
for the 50 non-stationary time series (right); b) PAE achieved per validation method across all 100 time series analyzed (left), only for the 50
stationary time series (center), only for the 50 non-stationary time series (right).

The median APAE across all 100 time series is lowest for bl-CV (0.03), followed by repOOS (0.05), and OOS, which

indicates the highest APAE (0.06) (Fig. 5a, left). The bl-CV therefore provides the most accurate estimate of the actual per-

formance of the 1D-CNN on the out-set across all 100 validated and tested GWL time series. The interquartile range (IQR),

which indicates the dispersion of the middle 50% of APAE, is the same for bl-CV and repOOS (0.06) and higher for OOS

(0.10). Accordingly, bl-CV and repOOS provide more robust performance estimates than OOS across the entire data set.

Figure 5a (left) shows significant upward outliers, indicating a strong discrepancy between $RMSE_{in}$ and $RMSE_{out}$ for some

time series. The associated analysis of the PAE values (Fig. 5b, left) shows that the upward outliers of the APAE values (Fig. 5a,

left) for bl-CV and repOOS predominantly represent strong underestimates of $RMSE_{out}$ (optimistic estimates), while for OOS

stronger overestimates (pessimistic estimates) occur. With regard to the median value of PAE across the entire data set (Fig.

5b, left), bl-CV with a median value of PAE of <0.01 shows no systematic bias in the performance estimate, while repOOS

(median PAE = 0.01) leads to slightly pessimistic performance estimates, and OOS (median PAE = 0.04) leads in median to

slightly overestimations of $RMSE_{out}$.



The analysis of APAE for stationary time series (Fig. 5a, center) reveals that the APAE values are lower in the median than in the entire data set (Fig. 5a, left). The bl-CV and repOOS methods have a lower median APAE (0.02) for the stationary time series, than the OOS approach (0.04). The PAE values for this group (Fig. 5b, center) are in the median <0.01 for bl-CV, 0.01 for repOOS, and 0.03 for OOS. The bl-CV indicates no systematic bias in performance estimation in the median, while repOOS and OOS demonstrate slightly pessimistic estimates. Notably, OOS in the stationary time series demonstrates the occurrence of upward outliers in APAE.

In Figure 5a (right), the APAE values are shown only for the non-stationary time series. It is noticeable that the APAE values of all three validation methods show higher median APAE values compared to the full data set (Fig. 5a, left). The median APAE for bl-CV and repOOS is 0.06, and for OOS, 0.08. Thus, the performance estimation of OOS for non-stationary time series is slightly worse than for bl-CV and repOOS. The scatter of APAE for this group of time series (Fig. 5a, right), represented by the IQR, is greater for all three validation methods compared to the stationary time series (Fig. 5a, center). In particular, OOS shows the greatest dispersion, with an IQR of 0.14 compared to bl-CV (IQR = 0.07) and repOOS (IQR = 0.06). In the non-stationary time series, all three validation methods show the presence of outliers with high APAE, a phenomenon also observed in the overall analysis (Fig. 5a, left). This finding indicates that non-stationary time series occasionally generate particularly high APAE. An examination of the PAE values for this group of time series (Fig. 5b, right) reveals that the extreme APAE outliers, for bl-CV and repOOS represent strong underestimations of $RMSE_{out}$, while for OOS, both strong underestimations and overestimations are observed. In the median, bl-CV achieves a PAE of <0.01, repOOS a PAE of -0.01, and OOS a PAE of 0.04. Consequently, bl-CV exhibits no systematic bias in the median performance assessment, repOOS provides marginally optimistic assessments, and OOS provides slightly pessimistic performance estimations.

Figure 6 illustrates the relationship between $RMSE_{in}$ and $RMSE_{out}$ for stationary (left) and non-stationary (right) time series. Each point shows the $RMSE_{in}$ and $RMSE_{out}$ of one time series. The distance of the points from the diagonal line ($RMSE_{in}$ = $RMSE_{out}$) indicates the resulting APAE. Points located above the diagonal reflect a positive deviation, indicating an overestimation of $RMSE_{out}$, while points below the diagonal reflect a negative deviation, indicating an underestimation of $RMSE_{out}$. A visual analysis of the scatter plot reveals that the points corresponding to OOS exhibit a greater spread around the diagonal compared to those of bl-CV and repOOS. This increased dispersion is also reflected by the larger IQR observed in the boxplots in Figure 5, both for stationary (Figure 5, center) and non-stationary (Figure 5, right) time series. The scatter plot for stationary time series shows that OOS estimates $RMSE_{out}$ less accurately (i.e., with greater deviation from the diagonal) and more frequently overestimates it (points above the diagonal) compared to bl-CV and repOOS. These time series display slightly abnormal GWL conditions during the OOS validation period in comparison to previous years—conditions that the model fails to predict accurately. In contrast, the GWL conditions in the out-set more closely resemble those observed prior to the OOS validation period.

An analysis of the scatter plot for the non-stationary time series in Figure 6 (right) shows that the prediction errors for non-stationary time series are higher than those for stationary time series in both in-set and out-set. As a result, the prediction uncertainty is considerably greater for non-stationary time series. Moreover, Figure 6 (right) illustrates that the spread of the data points across all validation methods is generally large and increases with higher $RMSE_{out}$. This suggests that the robustness



of performance estimates decreases as prediction errors increase, since larger APAE values reflect poorer model fit. The limited prediction accuracy and the resulting suboptimal performance estimates are most evident in non-stationary time series. This effect is attributable to the presence of trends or abrupt changes that cannot be adequately captured by the meteorological input sequences. A notable pattern is the tendency of OOS to overestimate high $\mathrm{RMSE_{out}}$, while bl-CV and repOOS tend to

underestimate them (see Figure 6, right).

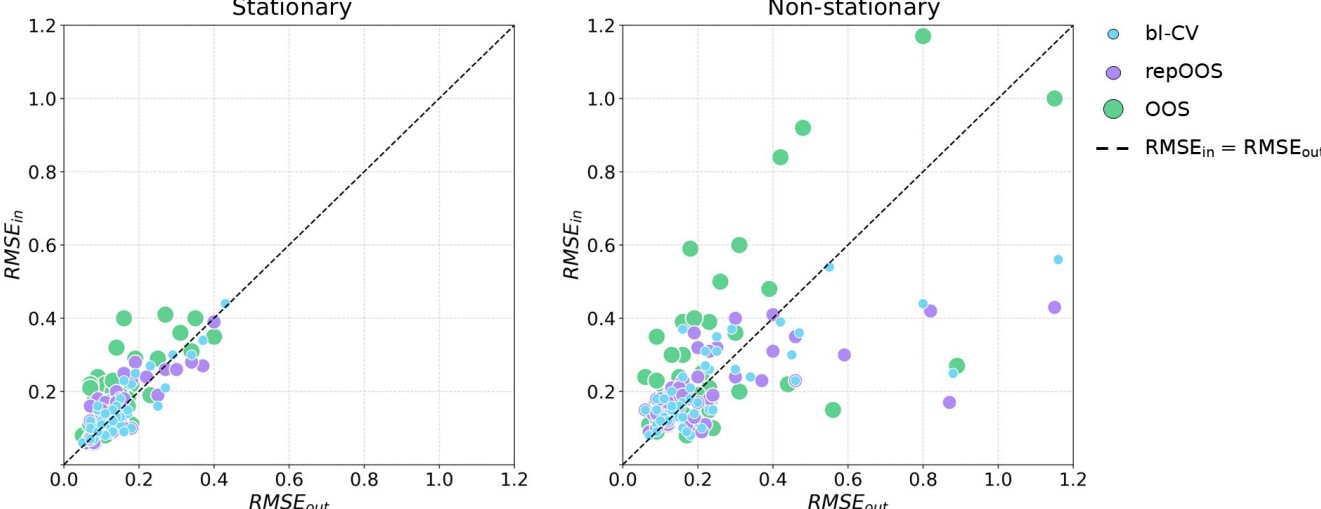

**Figure 6.** Point plot of the achieved $\mathrm{RMSE_{in}}$ and $\mathrm{RMSE_{out}}$. Each point represents the $\mathrm{RMSE_{in}}$ and the $\mathrm{RMSE_{out}}$ of a time series validated and tested with the corresponding validation method. Left: stationary time series; Right: non-stationary time series.

Figure 7 displays both the in-set predictions (left) with $\mathrm{RMSE_{in}}$ values in the titles and the out-set predictions (right) with the respective $\mathrm{RMSE_{out}}$ in the titles for the stationary time series 42. All validation methods achieve an RMSE of 0.07 in the out-set, indicating consistent model performance across validation strategies when evaluated on unseen data. The resulting $\mathrm{RMSE_{in}}$ amounts to 0.10 for bl-CV (averaged over all folds), 0.21 for OOS, and 0.12 for repOOS (averaged over all repetitions).

An analysis of the in-set predictions of bl-CV shows that for folds 1–4 (1991 to the beginning of 2010), the model predictions closely follow the observed groundwater levels. However, the 1D-CNN model tends to overestimate low groundwater levels. This changes in the subsequent period (fold 5, 2010 to the end of 2014), which corresponds to the OOS validation period (see Figure 7, center), higher groundwater levels occur more frequently and are underestimated by the model. Since the RMSE for bl-CV is calculated over all five folds, the influence of the deviations in fold 5 is reduced. This results in a more robust

performance estimate that is less sensitive to specific deviations. A similar effect is observed for repOOS, where components of repetitions 3 and 5 also lie within the 2010–2014 period (see Figure 7, bottom). However, the remaining repetitions cover different time spans, further reducing the influence of outlier periods. As a result, the average RMSE across all repetitions provides a more robust performance estimate. In contrast, OOS relies on a single validation period, making its performance estimate highly sensitive to local anomalies in the data.





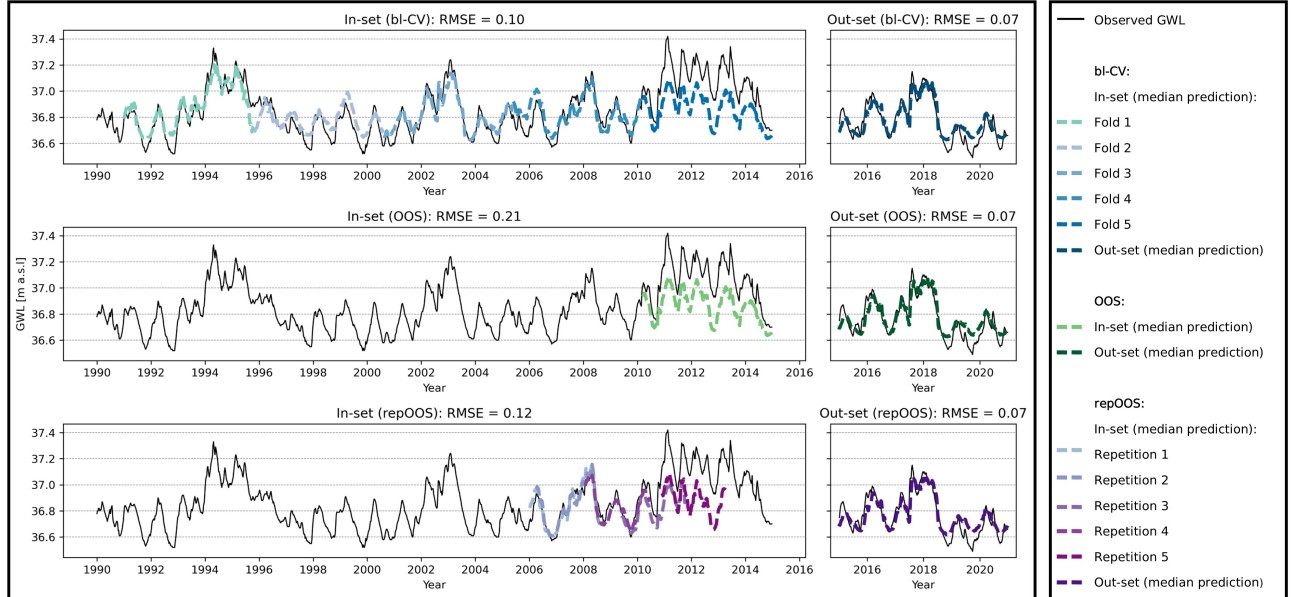

**Figure 7.** Plot of time series 42 (stationary time series, identified with ADF and KPSS). Top: validation forecasts of the 5-folds of the bl-CV (left) and test forecasts of the bl-CV model (right); Middle: validation forecast of the OOS validation period (left) and test predictions of the OOS model (right); Bottom: validation predictions of the repetitions of repOOS (left) and test predictions of the repOOS model (right).

An illustrative example of validation inconsistency is shown in Figure 8, where the model exhibits poor predictive performance and the validation methods either strongly over- or underestimate $\text{RMSE}_{\text{out}}$. This particular time series displays a steady increase in GWL from 2002, peaking in early 2014, followed by a decline during the out-set phase until the end of 2021. The meteorological input sequences provided to the 1D-CNN model do not capture the underlying trend dynamics in this case, resulting in high RMSE values in both the in-set and out-set. A detailed analysis reveals that $\text{RMSE}_{\text{out}}$ is significantly under-

estimated by both bl-CV and repOOS. This discrepancy arises because $\text{RMSE}_{\text{in}}$ is calculated as the average value over the periods with smaller and larger differences between predictions and observations, leading to an overly optimistic performance estimate. The validation period of the OOS, in contrast, coincides with a phase of rising groundwater levels and a peak in early 2014, during which the model substantially underestimates the actual GWL. Since this estimate is not averaged across periods of better performance, the resulting $\text{RMSE}_{\text{in}}$ of OOS reflects a more pessimistic assessment of model performance. Overall,

the suboptimal predictive capability of the model is evident in the $\text{RMSE}_{\text{in}}$ values of all three validation methods. However, the associated uncertainties in the performance estimates remain substantial.

     These findings underscore the need for particular care when evaluating non-stationary time series. In such cases, there is a heightened risk that the statistical properties (e.g., mean, variance) of the training, validation, and test periods differ significantly. These discrepancies complicate both the prediction and the evaluation processes, posing greater challenges than

stationary time series.





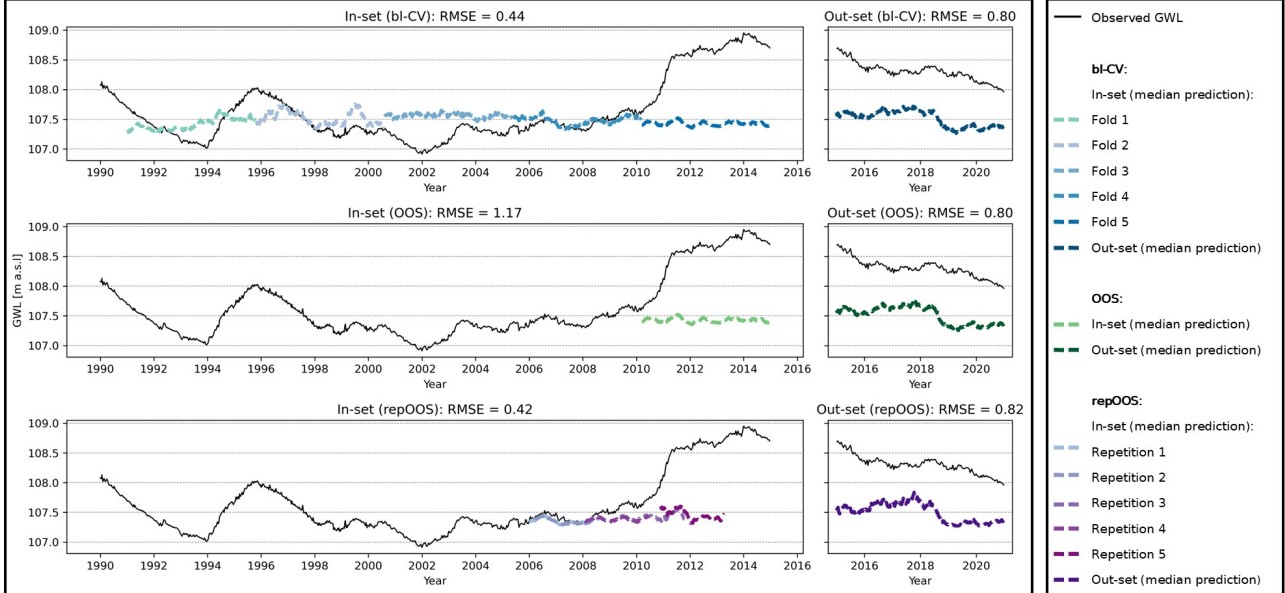

**Figure 8.** Plot of time series 97 (non-stationary time series, identified with ADF and KPSS). Top: validation forecasts of the 5-folds of the bl-CV (left) and test forecasts of the bl-CV model (right); Middle: validation forecast of the OOS validation period (left) and test predictions of the OOS model (right); Bottom: validation predictions of the repetitions of repOOS (left) and test predictions of the repOOS model (right).

## 5 Conclusion

The selection of an appropriate performance estimation method is a crucial step in the development and evaluation of predictive models. In hydrogeological modeling, the choice of a suitable validation approach remains an open challenge. In other domains, various studies have compared validation approaches such as CV or OOS with respect to their suitability for evaluating the

performance of autoregressive time series forecasts (e.g. Bergmeir and Benitez, 2011; Bergmeir et al., 2014, 2018; Cerqueira et al., 2017, 2020). These studies concluded that bl-CV yields the most robust and accurate performance estimates for stationary time series (Bergmeir and Benitez, 2011; Bergmeir and Benítez, 2012), while Cerqueira et al. (2020) recommended repOOS for autoregressive forecasting of non-stationary time series.

However, long-term forecasts are often required in hydrogeology, which necessitates a departure from autoregressive models

and instead favors the use of purely exogenous input data, such as meteorological forcings which are also available as climate forecasts. As a result, findings from earlier studies on autoregressive models may not be directly transferable to groundwater level (GWL) prediction based on exogenous input variables. This study addressed this gap by comparing three validation methods (bl-CV, repOOS, and OOS) for predicting 100 GWL time series (50 stationary and 50 non-stationary), using exogenous meteorological variables (precipitation, air temperature [mean, minimum, maximum], relative humidity, and a seasonal

temperature sinusoid). The GWL one week into the future was predicted using a 52-week input sequence of the meteoroligal



time series. The validation methods were compared with respect to their ability to estimate model performance by calculating APAE and PAE, following the approach of Bergmeir et al. (2018).

The results indicated that bl-CV provided the most accurate and robust performance estimates— in the overall analysis of all 100 time series and in the separate analysis of stationary and non-stationary series. The repOOS ranked second, while
OOS produced the least reliable and least accurate estimates. The robustness of bl-CV and repOOS can be attributed to their validation over multiple time periods, in contrast to the single-period validation used in OOS, which is more vulnerable to performance outliers. Although the present results are not directly comparable to studies on autoregressive forecasts, they support the conclusions of Bergmeir and Benitez (2011), Bergmeir and Benítez (2012), and Cerqueira et al. (2020) insofar as bl-CV appears to be a more suitable choice for stationary time series compared to repOOS and OOS. For non-stationary time
series, however, the findings diverge from those of Cerqueira et al. (2020), who identified repOOS as the most suitable method. In contrast, our study found that bl-CV not only outperformed OOS, but also yielded more accurate and robust performance estimates than repOOS for non-stationary time-series.

Moreover, the results suggested that predictions based on stationary GWL time series led to more robust forecasts and more reliable performance estimates. In contrast, non-stationary time series were associated with higher prediction errors and
increased uncertainty in the performance assessment of all evaluated validation methods. The Stationary GWL time series displayed less complex patterns, which could be more effectively captured by meteorological input variables and, consequently, more accurately predicted by the 1D-CNN model. Non-stationary time series, by contrast, exhibited more intricate, time-dependent structures, complicating both prediction and validation. These more challenging conditions demand particular caution when interpreting validation results. Even after observing test errors and final prediction performance, expert judgment
is required to assess whether the selected validation and test periods are truly representative. Choosing extended validation and test periods, as applied in bl-CV and repOOS, can mitigate the influence of anomalous periods and improve the representativeness of performance estimates — ultimately supporting more informed and resilient groundwater management decisions.

*Code availability.*  The code and data used to reproduce the models and the results of this study can be found at: https://doi.org/10.5281/zenodo.16419012 (Doll et al., 2025).

*Data availability.*  The groundwater data used in this study is an anonymized subset of the publicly available data from the Brandenburg State Office for the Environment on the water information platform (https://apw.brandenburg.de). The anonymized time series are also available at:https://doi.org/10.5281/zenodo.16419012 (Doll et al., 2025).

*Author contributions.*  F.D.: Conceptualization, methodology, software, formal analysis, investigation, visualization, writing - creation of the initial design; T.L.: Conceptualization, methodology, review and editing, supervision; M.W.: Data acquisition, data preparation, review and



editing; S.K.: Data acquisition, data preparation, review and editing; S.B.: Project management, participation in conceptual discussions, review and editing.

*Competing interests.* The authors declare that they have no conflict of interest.

*Acknowledgements.* This study was conducted using Python 3.8. The following Python libraries were applied for data preparation, implementation of the prediction model and visualization of the results: Tensorflow 2.9 (Abadi et al., 2015), keras 2.9 (Chollet, 2015), numpy

1.24.3 (Harris et al., 2020), pandas 2.0.3 (The pandas development team, 2023), scikit-learn 1.3.0 (Pedregosa et al., 2011), matplotlib 3.7.2 (Hunter, 2007), scipy 1.10.1 (Virtanen et al., 2020), statsmodels 0.14.1 (Seabold and Perktold, 2010), geopandas 0.13.2 (Bossche et al., 2023).

This work was developed as part of the project: KIMoDIs-KI-based monitoring, data management and information system for coupled prediction and early warning of groundwater low levels and salinization, which is funded by the Federal Ministry of Research, Technology

and Space (BMFTR) as a joint project under the funding code 02WGW1662B for the funding measure "LURCH: Sustainable Groundwater Management" as part of the federal program "Wasser:N". Wasser:N is part of the BMBF strategy "Research for Sustainability (FONA)".



# Appendix

**Summarized analysis of APAE and PAE:**

**Table A1.** Descriptive statistics for APAE and PAE (total, stationary, non-stationary)

| | Total (n=100) | | | Stationary (n=50) | | | Non-stationary (n=50) | | |
|---|---|---|---|---|---|---|---|---|---|
| | bl-CV | repOOS | OOS | bl-CV | repOOS | OOS | bl-CV | repOOS | OOS |
| **APAE** | | | | | | | | | |
| mean | 0.06 | 0.07 | 0.10 | 0.03 | 0.03 | 0.05 | 0.09 | 0.10 | 0.14 |
| std | 0.10 | 0.11 | 0.11 | 0.02 | 0.03 | 0.05 | 0.13 | 0.15 | 0.14 |
| min | 0.00 | 0.00 | 0.00 | 0.00 | 0.00 | 0.00 | 0.00 | 0.00 | 0.00 |
| 25% | 0.01 | 0.02 | 0.03 | 0.01 | 0.01 | 0.02 | 0.02 | 0.03 | 0.04 |
| 50% | 0.03 | 0.05 | 0.06 | 0.02 | 0.02 | 0.04 | 0.06 | 0.06 | 0.08 |
| 75% | 0.07 | 0.08 | 0.13 | 0.04 | 0.05 | 0.07 | 0.09 | 0.09 | 0.18 |
| max | 0.63 | 0.72 | 0.62 | 0.09 | 0.10 | 0.24 | 0.63 | 0.72 | 0.62 |
| **PAE** | | | | | | | | | |
| mean | -0.02 | -0.01 | 0.04 | 0.00 | 0.01 | 0.04 | -0.03 | -0.04 | 0.05 |
| std | 0.11 | 0.13 | 0.14 | 0.04 | 0.04 | 0.06 | 0.15 | 0.17 | 0.19 |
| min | -0.63 | -0.72 | -0.62 | -0.09 | -0.10 | -0.07 | -0.63 | -0.72 | -0.62 |
| 25% | -0.03 | -0.03 | -0.01 | -0.02 | -0.02 | 0.00 | -0.08 | -0.06 | -0.02 |
| 50% | 0.00 | 0.01 | 0.04 | 0.00 | 0.01 | 0.03 | 0.00 | 0.01 | 0.04 |
| 75% | 0.02 | 0.05 | 0.09 | 0.02 | 0.04 | 0.06 | 0.03 | 0.06 | 0.14 |
| max | 0.21 | 0.17 | 0.44 | 0.07 | 0.09 | 0.24 | 0.21 | 0.17 | 0.44 |





**Summarized analysis of RMSE in the in-set and out-set:**

**Table A2.** Descriptive statistics for RMSE (in-set and out-set, total, stationary, non-stationary)

| | Total (n=100) | | | Stationary (n=50) | | | Non-stationary (n=50) | | |
|---|---|---|---|---|---|---|---|---|---|
| | bl-CV | repOOS | OOS | bl-CV | repOOS | OOS | bl-CV | repOOS | OOS |
| **RMSE (in-set)** | | | | | | | | | |
| mean | 0.17 | 0.14 | 0.26 | 0.15 | 0.16 | 0.19 | 0.21 | 0.20 | 0.29 |
| std | 0.19 | 0.14 | 0.35 | 0.08 | 0.07 | 0.09 | 0.12 | 0.09 | 0.24 |
| min | 0.06 | 0.06 | 0.06 | 0.06 | 0.06 | 0.06 | 0.08 | 0.09 | 0.08 |
| 25% | 0.10 | 0.08 | 0.14 | 0.10 | 0.12 | 0.13 | 0.13 | 0.14 | 0.15 |
| 50% | 0.15 | 0.12 | 0.18 | 0.13 | 0.14 | 0.16 | 0.16 | 0.18 | 0.20 |
| 75% | 0.22 | 0.23 | 0.43 | 0.16 | 0.18 | 0.22 | 0.26 | 0.24 | 0.34 |
| max | 0.56 | 0.43 | 1.17 | 0.44 | 0.39 | 0.41 | 0.56 | 0.43 | 1.17 |
| **RMSE (out-set)** | | | | | | | | | |
| mean | 0.20 | 0.19 | 0.19 | 0.15 | 0.15 | 0.15 | 0.24 | 0.24 | 0.24 |
| std | 0.17 | 0.17 | 0.17 | 0.08 | 0.08 | 0.08 | 0.21 | 0.21 | 0.21 |
| min | 0.05 | 0.06 | 0.05 | 0.05 | 0.06 | 0.05 | 0.06 | 0.06 | 0.06 |
| 25% | 0.11 | 0.11 | 0.11 | 0.09 | 0.09 | 0.10 | 0.13 | 0.12 | 0.12 |
| 50% | 0.15 | 0.15 | 0.15 | 0.13 | 0.13 | 0.13 | 0.18 | 0.18 | 0.16 |
| 75% | 0.22 | 0.20 | 0.21 | 0.16 | 0.16 | 0.16 | 0.25 | 0.24 | 0.24 |
| max | 1.16 | 1.15 | 1.15 | 0.43 | 0.40 | 0.40 | 1.16 | 1.15 | 1.15 |



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



**1D-CNN** one-dimensional convolutional neural network

**ADF** augmented Dickey-Fuller Unit Root


**APAE** absolute predictive accuracy error

**bl-CV** blocked cross-validation

**CV** cross-validation

**GW** groundwater

**GWL** groundwater level


**IQR** interquartile range

**KPSS** Kwiatkowski-Phillips-Schmidt-Shin

**LfU** State Office for the Environment Brandenburg

**m a.s.l.** metres above sea leve

**ML** machine learning


**OOS** out-of-sample validation

**P** precipitation sum





**PAE** predictive accuracy error

**repOOS** repeated out-of-sample validation

**rH** relative humidity


**RMSE** root mean squared error

**T** mean air temperature

**Tmax** maximum air temperature

**Tmin** minimum air temperature

**Tsin** sinosoidal curve fitted to the mean air temperature
