# Peer review of "Validation Strategies for Deep Learning-Based Groundwater Level Time Series Prediction Using Exogenous Meteorological Input Features"

_EGUsphere, 2025_

## Author Comment (AC1)

**Review 1**

O. This study aims to examine performance evaluation methods for ML-based groundwater-level prediction. Based on the abstract, the main contributions of the study appear to be twofold: (1) the use of time-lagged meteorological variables together with non-time-lagged groundwater data for groundwater-level prediction, and (2) the application of different performance evaluation strategies, including blocked 5-fold cross-validation (bl-CV), repeated out-of-sample validation (repOOS), and out-of-sample validation (OOS). Having read through the manuscript, I find this to be a commendable effort that provides a thoughtful and well-executed case study on groundwater-level prediction.

We would like to thank you for your detailed, substantiated review and positive evaluation of our work. We are very pleased that you consider our study as thoughtful and well-executed and appreciate the objectives and relevance. Your constructive comments will help us in further improving the manuscript and clarifying any unclear aspects. Thank you once again for your valuable feedback and for the time you have invested in reviewing our manuscript.

1. However, my main concern—and hence the principal weakness of the study—is that, although the manuscript persuasively situates itself within the existing literature through comparisons and contrasts, its scope is constrained by the exclusive reliance on a 1-D CNN model. This raises the question of whether the conclusions regarding evaluation methods are fully justifiable, transferable, and robust across other ML/DL approaches. In addition, the absence of a benchmark comparison for the proposed approach further undermines the strength of the study.

Thank you very much for this valuable and differentiated comment. The aim of this study was not to compare different prediction models, but to systematically examine and compare different validation strategies — specifically, blocked 5-fold cross-validation (bl-CV), out-of-sample validation (OOS), and repeated out-of-sample validation (repOOS)—in terms of their suitability for performance evaluation in stationary and non-stationary groundwater level time series. For this purpose, a 1D-CNN model was deliberately chosen as a well-established method to be able to consider the methodological differences between the validation strategies in isolation. Nevertheless, we understand the concerns regarding the limitation to a single DL model and the potentially limited transferability of the results. We therefore applied the same validation strategies to a second well-established DL-architecture, an LSTM model and found that the conclusions of the paper were not affected. The results obtained with the LSTM model essentially corresponded to those obtained with the 1D-CNN. We will incorporate these results into the revised version of the manuscript to provide additional evidence of the robustness and transferability of our conclusions and hope this will alleviate the concerns.

2. I appreciate that the authors have already defined a clear research question for the current study. However, I would still encourage including more discussion of related research to better inform and engage GMD's broad and sophisticated readership. For example, it would be valuable to compare your findings with those of Shen et al. (2022), highlighting in what ways your results are consistent or divergent. In addition, while the manuscript presents a strong stance on the time-consecutive hypothesis, it would also be helpful to discuss the perspective proposed by Zhang et al. (2022), who emphasize the importance of distributional representativeness across train/validation/test sets and suggest that this consideration may reduce the need for k-fold cross-validation. Including such comparisons would strengthen the manuscript by situating your contribution more clearly within the broader context of current research.

Thank you very much for bringing these references to our attention. We agree that the inclusion of the work by Shen et al. (2022) and Zhang et al. (2022) significantly strengthens the classification of our results in the current scientific context. Both studies address key issues that also underlie our work—namely, the importance of the data split strategy and the representativeness of training, validation,

and test periods for model evaluation. While Shen et al. (2022) emphasizes that the choice of time division has a significant impact on the transferability of model results, Zhang et al. (2022) argues for the importance of a distribution-consistent selection of data sets. These two perspectives complement our investigation very well. We will expand the discussion in the revised manuscript version accordingly to highlight these connections more clearly and to relate our results to the aforementioned works.

3. Given that I do not have prior experience in groundwater-level modeling, I find that the opening paragraph of the introduction does not clearly convey the nature of the problem. It remains unclear whether the study is addressing point prediction, area-averaged prediction, or image-type prediction on structured or unstructured grids. In addition, it would be helpful to explain what the common problem setups have been in previous research along this line, so that readers without domain expertise can better situate the present study.

Thank you for this helpful comment. We will define the problem more clearly and explain in more detail in the manuscript that the predictions are time series predictions at measurement points. In addition, we will highlight the problems that frequently arise in previous work so that readers without specific expertise in groundwater modeling can better understand the motivation and context of the present study.

4. At the end of the Introduction, please provide a clear definition of what is meant by stationary and non-stationary conditions in the context of this study. Since these concepts can depend on the choice of window size, it would be helpful if you could explicitly state how they are defined here.

We agree that providing a clearer definition of 'stationary' and 'non-stationary' conditions in the context of our study would help readers to understand it more easily. Therefore, we will add an explicit description to the end of the introduction, explaining how these terms are used in our work. We will also clarify that the assessment of stationarity is based on analyzing the whole series, rather than using a moving window approach.

5. In the Theory and Background section, you discuss each evaluation method individually. However, some of this information was already mentioned in the Introduction. While the content itself is sound, I recommend further polishing to reduce redundancy and improve delineation and readability.

We acknowledge that there is some overlap between the 'Theory and Background' section and the introduction. In the revised version of the manuscript, we will revise this section specifically to reduce content redundancy, clarify the distinction between the introduction and theory sections, and improve the manuscript's structure and readability.

6. Around line 110, the manuscript states: "...For time series prediction, random shuffling of the data is often considered problematic as it can break the temporal dependency of the data...". I would like to clarify whether this claim is model-dependent. For instance, is this necessarily true when using tree-based algorithms such as Random Forests, which do not rely on temporal (Markovian) state updates? While sequential models that rely on temporal dependencies (e.g., autoregressive or state-space models) may indeed be affected, models that only map input—output relationships may not experience the same issue. Could you clarify whether this limitation arises primarily from the choice of model, rather than from the evaluation method itself?

Thank you for this helpful comment. You raise an important point regarding time series modelling that must be considered when interpreting our statement.

Generally, when making predictions based on time series or modelling the relationship between past and future data (e.g. predicting groundwater levels based on sequences of past meteorological

conditions), the order of the time steps should not be mixed at random, as this would destroy the temporal relationship between the input and the target value. This applies whether sequential models (e.g. autoregressive models or RNNs) or models such as random forests are used. Tree-based models also require temporally correct input-output pairs, for example if they use mean precipitation values from the last four, eight, or twelve weeks as features to predict groundwater levels at a future point in time.

However, once these input—output pairs have been constructed, they can be mixed within the training set, since the temporal relationship within an input vector, and between that input vector and the corresponding target value, has already been established.

Another crucial point is that, in real-world applications, a prediction model can only be trained using past data to predict the future. Therefore, the test dataset in particular should reflect the real application situation and be temporally after the training dataset to enable realistic evaluation of model performance.

In summary, the restriction on random mixing of time series affects not only the choice of model but is also a fundamental requirement for all prediction models that map time-dependent input-output relationships, whether they are random forests, CNNs or other models.

We will clarify this in the revised version of the manuscript.

7. Around line 175, you refer to the concept of weak stationarity. Could you please clarify what window size is being used to assess this property? Since the definition of weak stationarity can depend on the temporal window considered, this specification would help readers interpret the results correctly.

We used the Augmented Dickey-Fuller (ADF) test and the KPSS test to determine the stationarity of the time series, with both tests being calculated using the default settings from the statsmodels package.

Our aim was to establish whether the time series were stationary throughout the observation period, or if they exhibited long-term trends, drifts or structural changes. therefore, were therefore applied to the complete time series, as this enables a comprehensive evaluation of the stationary or non-stationary nature. A rolling window analysis was not planned here, as the focus was on long-term temporal changes rather than local ones.

In the revised version of the manuscript, we will clarify these methodological aspects and the definitions of stationary and non-stationary conditions to improve comprehensibility for readers.

8. For section 3.3, I wonder if the use of "dropout" would affect the results of using different evaluation method.

Dropout is a well-established method of regularization that reduces the risk of overfitting in artificial neural networks, thereby improving their ability to generalize. Dropout also improves robustness against noisy input data and generally leads to more stable training. Due to its many advantages, dropout is now used as standard in deep learning networks. From our conceptual understanding, we think it is unlikely that the use of dropout will fundamentally alter the results of this study.

9. If I understand correctly, the authors use 80% of the in-set data for model development. Have you evaluated whether a smaller subset of this 80% could achieve comparable accuracy and robustness, and, if so, what the minimum percentage might be? Additionally, would reducing the total amount of data alter the study's conclusions?

The question of whether a smaller proportion of the data could be used to develop a model with comparable accuracy and robustness is scientifically exciting and understandable. In the present study, we opted for 80/20 split, which is commonly employed in groundwater level prediction (Ahmadi et al., 2022) as well as in many other disciplines. This approach also enables better comparability with other studies that employed data splits of 70/30 (e.g. Bergmeir et al., 2018; Cerqueira et al., 2017) or 80/20 (e.g. Bergmeir & Benítez, 2012). Our aim was to focus clearly on comparing the validation strategies, rather than expanding the scope of the study by assessing how additional variations in data splitting affect the results.

However, analyses with different data splits could provide valuable insights into the stability of the models and the transferability of the results and therefore represent an interesting approach for future research.

10. Figures 5–7 provide a reasonable and effective way of summarizing the results. That said, are there additional quantitative approaches that could be used to present the findings on spatial maps? Moreover, beyond the stationarity perspective, could further insights be derived in terms of predictive accuracy that would enrich the interpretation of the results?

Thank you for this constructive comment. We acknowledge that additional quantitative approaches, such as spatial analysis, could provide further interesting insights. However, we would like to point out that the focus of this study was deliberately on the methodological aspect of correctly evaluating the performance of the validation strategies, rather than on the predictive accuracy or hydrogeological interpretability of the model. Therefore, to keep the objective clear and focused, we would like to refrain from making more far-reaching interpretations of the results, e.g. regarding their spatial distribution.

11. The manuscript fixes the input meteorological sequence length at 52 weeks. Please clarify the basis for this choice. Have alternative horizons been tested? Should the optimal horizon be constant across sites, or might it vary with hydro-geo-climatic setting? If not constant, what insights can be derived from treating this as a site-specific (or region-specific) hyperparameter?

The proposal to consider the length of the meteorological input sequence as a potentially location-dependent hyperparameter is sensible and could provide additional interesting insights.

However, since the study focused on evaluating validation approaches rather than optimizing model architecture, we used a uniform hyperparameter setting. This allowed us to ensure the comparability of results across the various validation strategies. The length of the meteorological input sequence was set to 52 weeks to map a complete annual cycle and thus fully account for seasonal influences. For this reason, the sequence length of 52 weeks was also chosen in other studies in the groundwater sector (e.g. Kunz et al., 2025; Wunsch et al., 2024)

Nevertheless, we agree that investigating variable or site-specific input sequences in future work could provide valuable insights into how hydro-geo-climatic differences influence model performance.

**References**

- Ahmadi, A., Olyaei, M., Heydari, Z., Emami, M., Zeynolabedin, A., Ghomlaghi, A., Daccache, A., Fogg, G. E., & Sadegh, M. (2022). Groundwater Level Modeling with Machine Learning: A Systematic Review and Meta-Analysis. *Water*, *14*(6), Article 6. https://doi.org/10.3390/w14060949
- Bergmeir, C., & Benítez, J. M. (2012). On the use of cross-validation for time series predictor evaluation. *Information Sciences*, 191, 192–213. https://doi.org/10.1016/j.ins.2011.12.028
- Bergmeir, C., Hyndman, R. J., & Koo, B. (2018). A note on the validity of cross-validation for evaluating autoregressive time series prediction. *Computational Statistics & Data Analysis*, 120, 70–83. https://doi.org/10.1016/j.csda.2017.11.003
- Cerqueira, V., Torgo, L., Smailović, J., & Mozetič, I. (2017). A Comparative Study of Performance Estimation Methods for Time Series Forecasting. *2017 IEEE International Conference on Data Science and Advanced Analytics (DSAA)*, 529–538. https://doi.org/10.1109/DSAA.2017.7
- Kunz, S., Schulz, A., Wetzel, M., Nölscher, M., Chiaburu, T., Biessmann, F., & Broda, S. (2025).

  Towards a global spatial machine learning model for seasonal groundwater level predictions in Germany. *Hydrology and Earth System Sciences*, *29*(15), 3405–3433.

  https://doi.org/10.5194/hess-29-3405-2025
- Wunsch, A., Liesch, T., & Goldscheider, N. (2024). Towards understanding the influence of seasons on low-groundwater periods based on explainable machine learning. *Hydrology and Earth System Sciences*, 28(9), 2167–2178. https://doi.org/10.5194/hess-28-2167-2024

---

## Author Comment (AC2)

**Review 2**

I Cannot find the importance of the presents study and how it contributes to the improvement of our knowledge in terms of GWL prediction using machine learning. Predicting GWL using one deep learning model (the CNN) is not new and the fact that the authors propose a modelling strategy based only on exogenous variables is no as important to be proposed and presented as an innovative approach. Furthermore, the fact that three different evaluation strategy, i.e., blocked cross-validation (5 bl-CV), repeated out-of-sample validation (repOOS), and out-of-sample validation (OOS) are compared is not a solid argument to justify the importance and novelty of the present paper. Yet, a modelling strategy based only on one ML model is extremely unsound as there is no any baseline of comparison. The adoption of weekly data is not justified and for closing, section results is extremely poor and unsound. There is no any interpretability of the model and a ranking of the features based on their contribution to the final model response.

Despite the harsh criticism, which we believe is neither substantiated nor justified, we would like to thank the reviewer for their thorough review of our manuscript. We regret that the significance of our study was not fully apparent to them, and we will use this opportunity to provide further clarification in the revised version of the manuscript.

Nevertheless, we would like to clarify a few key points below.

- First, we would like to emphasize that our study did not aim to present the use of exogenous variables or a CNN for prediction as innovative approaches, nor did it aim to compare multiple model architectures. In fact, there is already a wealth of research on these aspects. Similarly, aspects such as model interpretability and feature ranking were not emphasized, as the methodological focus was on the performance evaluation of the validation strategies.
- We aimed to investigate how different validation strategies blocked cross-validation (bl-CV), repeated out-of-sample validation (repOOS) and classic out-of-sample validation (OOS) influence the accuracy of performance evaluation. We are very sorry that Reviewer 2 does not consider this sufficient to justify the importance and novelty of the present paper. Various publications in other disciplines have recognized that validation strategies do indeed have a significant influence (e.g. Bergmeir et al., 2014, 2018; Bergmeir & Benitez, 2011; Bergmeir & Benítez, 2012; Cerqueira et al., 2020). We therefore consider it legitimate to investigate whether and to what extent these findings can be transferred to the field of groundwater level prediction. The main relevance of our work lies in our critical analysis of the effects of data splitting strategies and the selection of representative time periods for training, validation and testing. This methodological issue is of great practical and scientific importance, particularly in the prediction of groundwater levels, where time series are frequently non-stationary and exhibit long-term trends or seasonal effects. Incorrectly selecting training or test periods can lead to misleading model evaluations, an issue that is often overlooked in many machine learning applications.
- Regarding the criticism of using a single model (1D-CNN), we would like to clarify that our focus is on comparing validation strategies, not innovating models. This is why an established model architecture was chosen. As we explained in our response to Reviewer 1, the findings on the robustness and transferability of the evaluation methods can be applied to other ML approaches. However, to strengthen the results further, we applied validation methods to an LSTM model. We will include these results, which hardly differ from those of the 1D-CNN, in the revised version of the manuscript.

 Weekly data is commonly used in groundwater level prediction as it accurately represents the dynamics of most aquifers and strikes a good balance between data availability and the volume of training data required. We will take this into account in the revised version of the manuscript.

We hope that these explanations clarify the significance and relevance of our study. Our work provides practical guidance regarding the selection of training, validation and test data periods in ML models for groundwater level forecasting. Our work expands on existing research approaches, which have primarily focused on autoregressive models to date.

**References**

- Bergmeir, C., & Benitez, J. M. (2011). Forecaster performance evaluation with cross-validation and variants. *2011 11th International Conference on Intelligent Systems Design and Applications*, 849–854. https://doi.org/10.1109/ISDA.2011.6121763
- Bergmeir, C., & Benítez, J. M. (2012). On the use of cross-validation for time series predictor evaluation. *Information Sciences*, 191, 192–213. https://doi.org/10.1016/j.ins.2011.12.028
- Bergmeir, C., Costantini, M., & Benítez, J. M. (2014). On the usefulness of cross-validation for directional forecast evaluation. *Computational Statistics & Data Analysis*, 76, 132–143. https://doi.org/10.1016/j.csda.2014.02.001
- Bergmeir, C., Hyndman, R. J., & Koo, B. (2018). A note on the validity of cross-validation for evaluating autoregressive time series prediction. *Computational Statistics & Data Analysis*, 120, 70–83. https://doi.org/10.1016/j.csda.2017.11.003
- Cerqueira, V., Torgo, L., & Mozetič, I. (2020). Evaluating time series forecasting models: An empirical study on performance estimation methods. *Machine Learning*, *109*(11), 1997–2028. https://doi.org/10.1007/s10994-020-05910-7

---

## Author Comment (AC3)

**Review 3**

This paper reviews and analyzes the validation strategies of time series deep learning models. They classify the metric performance assessment approaches as three groups: out-of-sample validation (OOS), blocked cross-validation (bl-CV), and repeated out-of-sample validation (repOOS). Subsequently, they establish one Dimension Convolutional Neural Networks (1D-CNN) model considering exogenous meteorological inputs with time lags for groundwater level (GWL) prediction. And then, a data set of 100 GWL time series (including 50 stationary and 50 nonstationary time series) in Brandenburg, Germany, are used to assess the validation strategies. Finally, they confirm that bl-CV and repOOS provide the most representative performance estimates for stationary and nonstationary GWL data, respectively. This paper more likes a review of validation strategies, and lacks deep analysis, especially for hydrogeological conditions.

Thank you for your constructive review of our manuscript and your valuable comments. We would now like to address each point individually:

1. Please clarify the contribution of this paper to hydrology.

The focus of our study is to systematically investigate the accuracy of the performance evaluation of validation strategies for machine learning (ML) models. We aimed to stimulate critical examination of the selection of data splits and time periods for training, validation and test data, particularly in the case of non-stationary time series, for which our results are highly relevant. Our work builds on existing studies in other fields that have focused primarily on autoregressive models by examining a non-autoregressive approach and demonstrating how different validation strategies influence performance estimation. We think that these considerations are especially important in hydrology (but also in other geoscientific disciplines) because of the often non-stationary nature of the hydrological (or in general environmental) time series. We will further clarify this in the revised version of the manuscript.

2. Did you consider alteration of the loss functions in the training period to identify suitable hyper parameters for improving the model performance?

To focus methodologically on comparing the validation strategies, we deliberately refrained from further hyperparameter optimization, including adjustment of the loss function. Only the number of training epochs was adjusted to ensure stable training without overfitting. Further investigation of comprehensive hyperparameter optimization could be considered in future work. However, it should be noted that hyperparameter optimization naturally depends on model performance during the selected validation period, which is influenced by the choice of this period. This is a problem in which everything is interrelated. This made it even more important to consider the validation strategies in isolation first, keeping all other parameters constant as far as possible.

3. The better actual model performance of GWL should not only have the small APAE, but also reflect the heterogeneity of aquifers (e.g., response time of GWL to meteorological factors, and amplitude).

We agree that factors such as response times to meteorological influences and the amplitude of groundwater levels are important for interpreting the performance of the model. However, the focus of this study was on the comparative evaluation of validation strategies, which is why standardized performance metrics such as APAE were used.

4. A systematical analysis the hydrogeological conditions of study area are needed (e.g., how many layers of aquifers, which layers the 100 GWL wells located, and groundwater pumping rates), which can help us figure out the best performance model.

A detailed analysis of hydrogeological parameters (e.g. the number of aquifers, the locations of wells and the rates of groundwater extraction) could further enrich the interpretation of model performance. However, the methodological focus of this work was not on achieving the best performance model, but to compare validation strategies using a representative sample of 100 random stationary and non-stationary time series.

The manuscript was submitted to Geoscientific Model Development rather than a purely hydro(geo)logical journal since our study is methodological in nature, examining the performance evaluation of time series models — a topic that may be relevant to, and transferable across, other geoscientific disciplines beyond hydrogeology. Therefore, the insights gained regarding the selection of training, validation and test data splits, as well as robust performance evaluation, are relevant not only for groundwater hydrology, but also for time series-based machine learning (ML) models in geoscientific applications.

In summary, this study makes an important methodological contribution by providing practical guidance on how to reliably evaluate the performance of machine learning (ML) models for time series data, particularly non-stationary time series. The study highlights the need to critically examine the selected validation periods. It shows that the length and location of the validation period within the time series both play a role in correctly assessing performance. This explains why bl-CV and repOOS outperformed the common OOS validation in our study. Thus, this work contributes to a more critical and reflective approach to validation strategies, complementing existing studies that have primarily focused on autoregressive models rather than striving for the best possible predictive accuracy and interpretability. We will further clarify this in the revised version of the manuscript.